# An acetylation-mediated chromatin switch governs H3K4 methylation read-write capability

Kanishk Jain[1,2], Matthew R Marunde[3], Jonathan M Burg[3], Susan L Gloor[3], Faith M Joseph[4], Karl F Poncha[4], Zachary B Gillespie[3], Keli L Rodriguez[3], Irina K Popova[3], Nathan W Hall[3], Anup Vaidya[3], Sarah A Howard[3], Hailey F Taylor[3], Laylo Mukhsinova[3], Ugochi C Onuoha[3], Emily F Patteson[3], Spencer W Cooke[1], Bethany C Taylor[4], Ellen N Weinzapfel[3], Marcus A Cheek[3], Matthew J Meiners[3], Geoffrey C Fox[5], Kevin EW Namitz[6], Martis W Cowles[3], Krzysztof Krajewski[1], Zu-Wen Sun[3], Michael S Cosgrove[7], Nicolas L Young[4], Michael-Christopher Keogh[3]*, Brian D Strahl[1,2,5]*

[1]Department of Biochemistry and Biophysics, University of North Carolina at Chapel Hill of Medicine, Chapel Hill, United States; [2]Lineberger Comprehensive Cancer Center, University of North Carolina at Chapel Hill, School of Medicine, Chapel Hill, United States; [3]EpiCypher, Inc, Durham, United States; [4]Verna & Marrs McLean Department of Biochemistry and Molecular Biology, Baylor College of Medicine, Houston, United States; [5]Curriculum in Genetics and Molecular Biology, University of North Carolina at Chapel Hill, School of Medicine, Chapel Hill, United States; [6]Pennsylvania State University, State College, United States; [7]Department of Biochemistry and Molecular Biology, Upstate Medical University, Syracuse, United States

*For correspondence:
mkeogh@epicypher.com (M-CK);
brian_strahl@med.unc.edu (BDS)

**Abstract** In nucleosomes, histone N-terminal tails exist in dynamic equilibrium between free/accessible and collapsed/DNA-bound states. The latter state is expected to impact histone N-termini availability to the epigenetic machinery. Notably, H3 tail acetylation (e.g. K9ac, K14ac, K18ac) is linked to increased H3K4me3 engagement by the BPTF PHD finger, but it is unknown if this mechanism has a broader extension. Here, we show that H3 tail acetylation promotes nucleosomal accessibility to other H3K4 methyl readers, and importantly, extends to H3K4 writers, notably methyltransferase MLL1. This regulation is not observed on peptide substrates yet occurs on the *cis* H3 tail, as determined with fully-defined heterotypic nucleosomes. In vivo, H3 tail acetylation is directly and dynamically coupled with *cis* H3K4 methylation levels. Together, these observations reveal an acetylation 'chromatin switch' on the H3 tail that modulates read-write accessibility in nucleosomes and resolves the long-standing question of why H3K4me3 levels are coupled with H3 acetylation.

## Editor's evaluation

This is a fundamental study revealing that cis H3 tail acetylation promotes nucleosome accessibility to histone methyl readers and writers. The findings are supported by compelling evidence providing an explanation for long-observed mechanisms of chromatin crosstalk.

## Introduction

In the epigenetic landscape, histone proteins are often variably chemically modified by 'writer' enzymes (*Jenuwein and Allis, 2001*; *Strahl and Allis, 2000*). Writer-installed post-translational modifications (PTMs) can then be recognized by 'reader' proteins and/or removed by 'eraser' enzymes. This interplay of PTMs comprises the 'histone code,' and has a central function in regulating chromatin organization and activity. For example, methylated/acylated lysine or methylated arginine residues of histones can recruit transcription factors to activate or repress transcription (*Strahl and Allis, 2000*; *Su and Denu, 2016*); mitotically phosphorylated serine/threonine residues can regulate reader binding established at earlier stages of the cell cycle (*Rossetto et al., 2012*) or ubiquitinated lysine can impact the maintenance of DNA methylation (*Vaughan et al., 2021*). As the complex language of histone PTMs is dissected, it has become clear that multivalent interactions with reader proteins can influence chromatin structure and DNA accessibility, thereby regulating gene transcription and other DNA-templated events (*Su and Denu, 2016*; *Taylor and Young, 2021*; *Young et al., 2010*). In this manner, combinatorial PTMs can more effectively engage different chromatin-binding modules, promoting distinct outcomes versus either PTM alone.

The bulk of chromatin PTM research has employed histone peptides, even though histones exist in vivo in a heteromeric complex with DNA (i.e. the nucleosome). Recent work, however, is making it increasingly clear that studying histone PTM engagement in the nucleosome context provides a more accurate understanding of the histone code. Particularly, the highly charged histone tails interact directly with nucleosomal DNA, restricting access to PTM recognition by reader proteins (*Ghoneim et al., 2021*; *Marunde et al., 2022a*; *Morrison et al., 2018*). Studies with BPTF PHD suggest acetylation releases the H3 N-terminal tail from the nucleosome surface, such that H3K4me3 becomes more readily engaged by the PHD finger (*Marunde et al., 2022a*; *Morrison et al., 2018*).

Considerable research effort has focused on dissecting the direct (and multivalent) engagement of chromatin via histone PTM-reader protein interactions. However, less appreciated are any indirect effects of PTMs on histone tail accessibility/nucleosome dynamics (e.g. via charge neutralization). In this report, we demonstrate enhanced nucleosome binding by a range of H3K4 readers when the histone tail is concomitantly acetylated (one or more of K9ac, K14ac, and K18ac). Furthermore, from in vitro enzymatic assays, we found that neighboring acetylation of the *cis* H3 tail is a prerequisite switch that enables the MLL1 complex (MLL1C: MLL1 SET domain, WDR5, RbBP5, Ash2L, and DPY30) (*Rao and Dou, 2015*) to robustly methylate H3K4. Consistent with this observation, mass spectrometric proteomic analyses of mammalian cells in a timed response to sodium butyrate (a broad-spectrum lysine deacetylase (KDAC) inhibitor) revealed a tight correlation of H3K4 methylation with *cis* acetylation. Our findings define a critical aspect of chromatin regulation: i.e., PTM cross-talk through acetylation-mediated tail accessibility. The findings also provide a molecular basis for the long-standing connection between H3K4 methylation and H3 acetylation in multiple eukaryotes (*Garcia et al., 2007*; *Nightingale et al., 2007*; *Taverna et al., 2007*), and resolve the directionality of these correlations: *cis* hyperacetylation of the H3 tail precedes, and is largely a prerequisite for, H3K4 methylation. Thus, the establishment of sites of H3K4me3 and activation of transcription occurs by a sequence of modifications of the same histone molecule.

## Results

### PHD finger readers show narrowed selectivity for histone tail PTMs on mononucleosomes versus peptides

How histone readers engage nucleosomes is an extensively researched area of chromatin biology. Most investigators characterize reader binding with PTM-defined histone peptides, although the domains often display a refined preference for similarly modified nucleosomes (*Marunde et al., 2022b*; *Marunde et al., 2022a*; *Morgan et al., 2021*; *Morrison et al., 2018*). To further assess this potential, we used the dCypher approach (*Jain et al., 2020*; *Marunde et al., 2022b*; *Morgan et al., 2021*; *Weinberg et al., 2021*) to measure the interactions of three PHD readers (from KDM7A, DIDO1, and MLL5: the queries) with PTM-defined peptides and nucleosomes (the potential targets). As might be expected (*Jain et al., 2020*), each GST-PHD fusion showed a preference for H3K4-methylated peptides and, particularly, for higher methyl states (i.e. KDM7A bound me2/me3, DIDO1 bound me1/me2/me3, and MLL1 bound me1/me2/me3; *Figure 1A*). In contrast, each GST-PHD reader

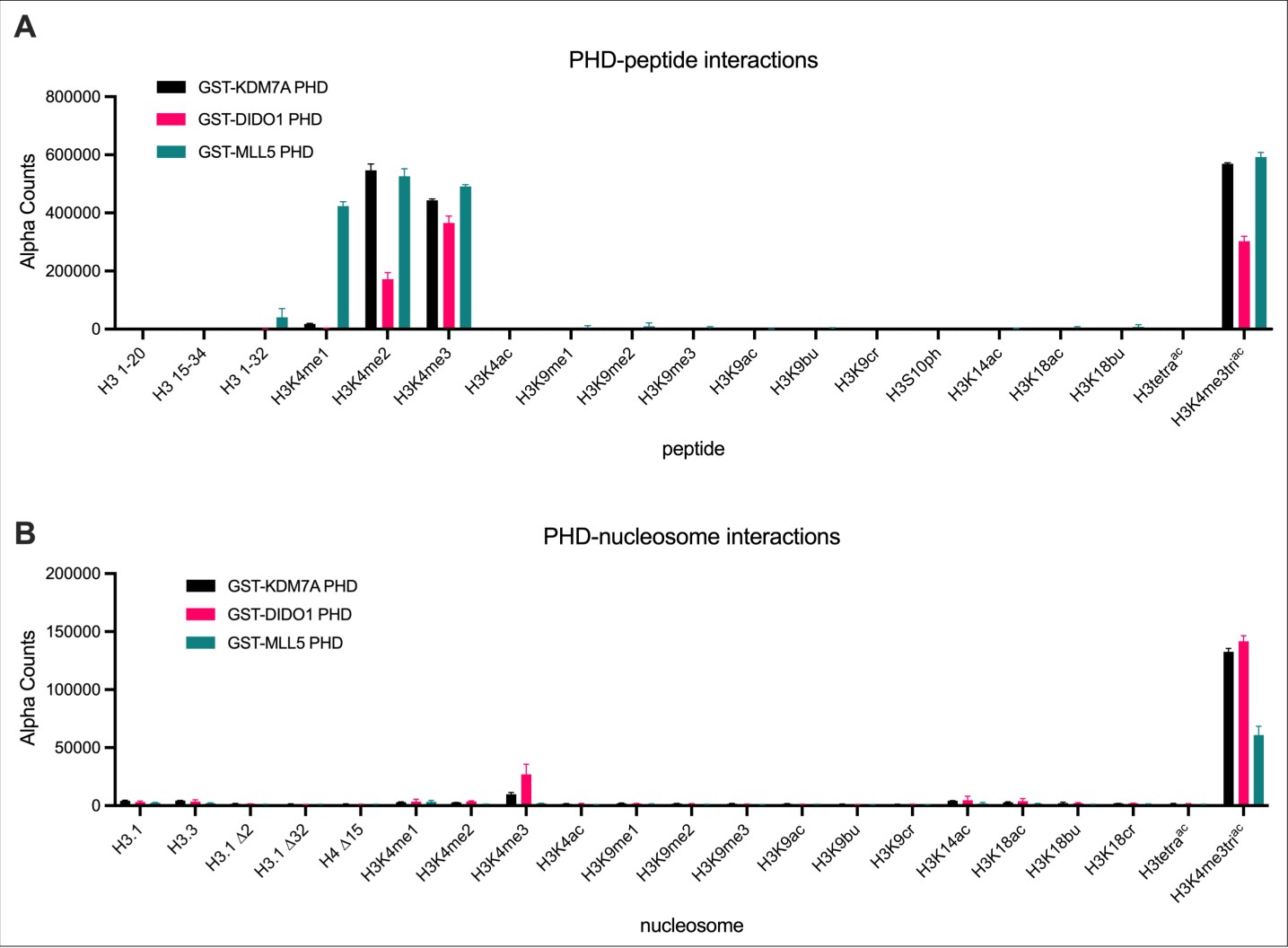

**Figure 1.** PHD finger reader domains show restricted binding on post-translational modification (PTM)-defined peptides vs. nucleosomes. *dCypher* assay alpha counts for the interaction of GST-PHD queries (9.5 nM KDM7A (Uniprot #Q6ZMT4; residues 1–100); 2.4 nM DIDO1 (Uniprot #Q9BTC0; residues 250–340); 18 nM MLL5 (Uniprot #Q8IZD2; residues 100–180)) with PTM-defined peptides (**A**) *vs.* nucleosomes (**B**) (the potential targets). All error bars represent the range of two replicates. Key: H3.1 Δ2, H3.1 Δ32, and H4 Δ15 are nucleosomes assembled with histones lacking the indicated N-terminal residues of H3.1 or H4. All data were plotted using GraphPad Prism 9.0. Of note, each reader query showed minimal interaction-free DNA (147 bp or 199 bp: *Figure 1—figure supplement 1B*).

The online version of this article includes the following source data and figure supplement(s) for figure 1:

**Figure supplement 1.** dCypher assays with PHD finger reader domains.

**Figure supplement 1—source data 1.** $EC_{50}^{Rel}$ binding values (in nM) for GST-PHD proteins with histone peptides and nucleosomes.

was restricted to H3K4me3 over the lower methyl states on nucleosomes (*Figure 1B*) and displayed weaker relative binding ($EC_{50}^{rel}$: calculated as in **Methods**) to this PTM (*Figure 1—figure supplement 1C–D*). On further examination, we observed no impact of co-incident acetylation on H3K4me3 binding in the peptide context (*Figure 1A*: compare H3K4me3 to H3K4me3K9acK14acK18ac [hereafter H3K4me3tri$^{ac}$]). In stark contrast, the binding of each GST-PHD reader to nucleosomal H3K4me3 was dramatically enhanced (~10–15 fold) by co-incident acetylation (i.e. H3K4me3tri$^{ac}$; *Figure 1B* and *Figure 1—figure supplement 1A–B*). Additionally, there was no significant reader domain interaction with either nucleosome lacking H3K4 methylation or 147X601 DNA (nucleosomal DNA) alone (*Figure 1—figure supplement 1B*). A similar observation has also been made for the BPTF PHD domain (*Marunde et al., 2022a*; *Morrison et al., 2018*), suggesting the potential for a general

mechanism. This led us to consider the possibility that histone tail lysine acetylation (Kac) might function beyond the recruitment of readers, and perhaps also impact H3K4 writers.

## H3 N-terminal acetylation enhances MLL1C-mediated methylation of nucleosomal H3K4

To investigate if acetylation might enable a more catalytically accessible H3 N-terminus, we performed enzymatic assays with the MLL1 core complex (MLL1C; responsible for H3K4me3) (*Rao and Dou,*

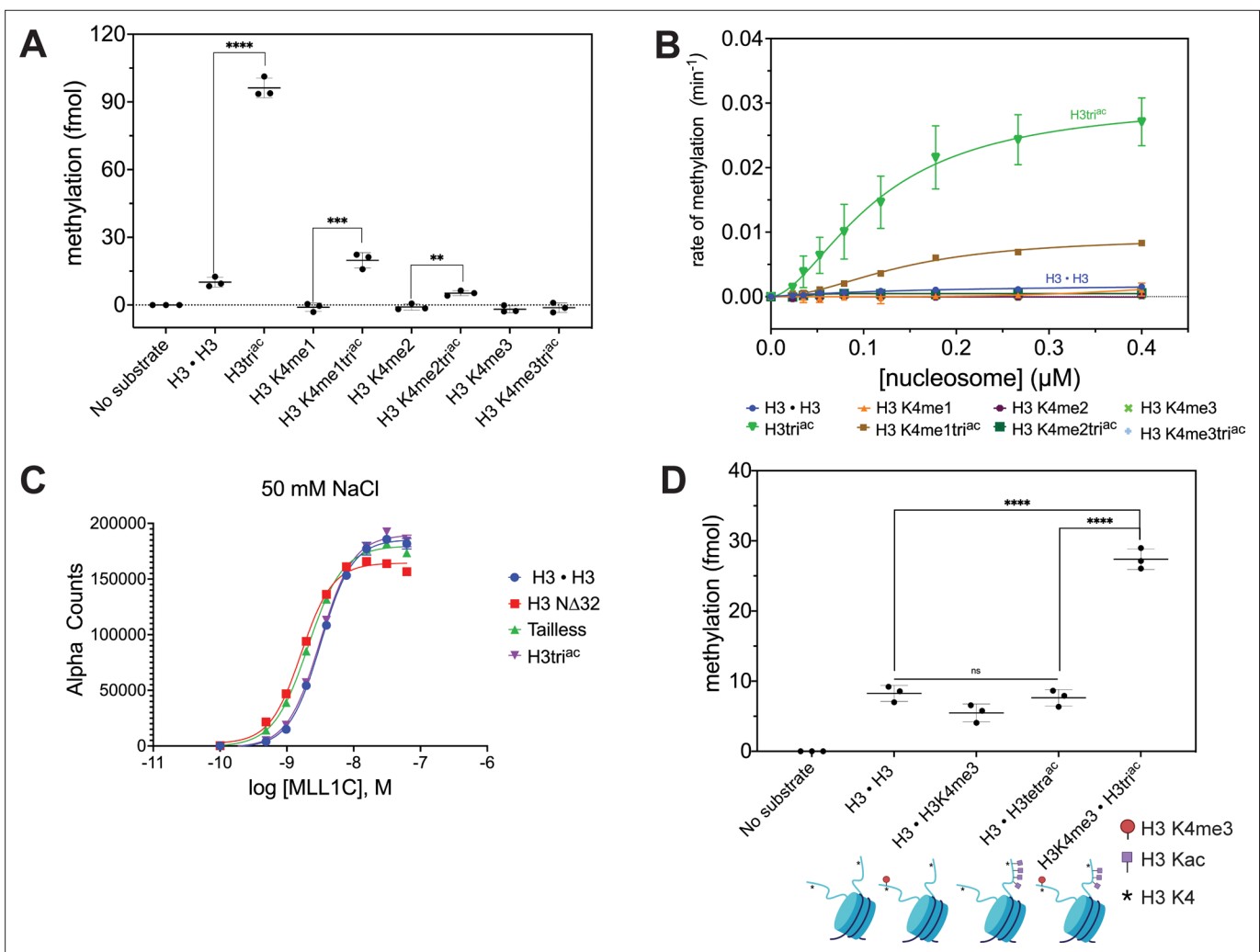

**Figure 2.** MLL1 complex (MLL1C) methylation activity on nucleosomal H3K4 is significantly enhanced by co-incident H3tri[ac]. (**A**) Endpoint methylation assays of H3K4me0 [H3 • H3]-me1-me2-me3 nucleosomes and their cognate H3K9ac14ac18ac (H3tri[ac]) partners (all 100 nM) with MLL1 Complex (MLL1C; 4 nM). Reactions performed in triplicate with error bars as SEM. *p*-values were determined using a two-tailed t-test: ****=<0.0001, ***=0.0008, **=0.0038. (**B**) MLL1C (4 nM) methylation activity on H3K4me0-me1-me2-me3 nucleosomes and their cognate H3tri[ac] partners (all substrates: 1.5-fold serial dilution, 0–0.4 μM). Reactions performed in triplicate with error bars as SEM (see also *Figure 2—figure supplement 1*). (**C**) MLL1C does not differentially associate with post-translational modification (PTM)-defined nucleosome substrates under study conditions. dCypher binding curves of hexahistidine (6HIS)-tagged MLL1C (concentrations noted) with PTM-defined nucleosomes (20 nM). Error bars represent the range of two replicates. (**D**) MLL1C-mediated methylation is enhanced on *cis* but not *trans*-acetylated nucleosomal H3 tails. Endpoint methylation assays of MLL1C (4 nM) with homotypic [H3 • H3] *vs.* heterotypic (e.g., [H3 • H3tetra[ac]]; see **Methods**) nucleosome substrates (all 100 nM). Reactions performed in triplicate with error bars as SEM. *p*-values were determined using a two-tailed t-test: ****=<0.0001. Key: H3tri[ac] = H3K9acK14acK18ac. H3tetra[ac] = H3K4acK9acK14acK18ac.

The online version of this article includes the following source data and figure supplement(s) for figure 2:

**Figure supplement 1.** Methylation assays with MLL1 complex (MLL1C) and nucleosome substrate.

**Figure supplement 1—source data 1.** Files of raw Coomassie gel images are used in *Figure 2—figure supplement 1*.

**Figure supplement 1—source data 2.** EC$_{50}^{rel}$ binding values (in nM) for MLL1C and nucleosomes.

**Table 1.** Steady-state Hill kinetic parameters.

| Substrate | $K_{0.5}$ (µM) | $k_{cat}$ (min⁻¹) | h (Hill coefficient) | $R^2$ |
|---|---|---|---|---|
| H3 • H3 | 0.13 ± 0.06 | 0.0018 ± 0.0004 | 1.213 ± 0.3430 | 0.8810 |
| H3 K9ac/14ac/18ac | 0.12 ± 0.02 | 0.0304 ± 0.0036 | 1.741 ± 0.3528 | 0.9228 |
| H3K4me1 | n.d.* | n.d.* | n.d.* | 0.3186 |
| H3K4me1 K9ac/14ac/18ac | 0.14 ± 0.008 | 0.0092 ± 0.0004 | 2.032 ± 0.1333 | 0.9920 |
| H3K4me2 | n.d.* | n.d.* | n.d.* | –0.1746 |
| H3K4me2 K9ac/14ac/18ac | 0.042 ± 0.005 | 0.0005 ± 0.00004 | 5.349 ± 2.682 | 0.8155 |
| H3K4me3 | n.d.* | n.d.* | n.d.* | n.d.* |
| H3K4me3 K9ac/14ac/18ac | n.d.* | n.d.* | n.d.* | n.d.* |

*= methylation signal was indistinguishable from the background: kinetic parameters could not be determined.

*2015*; *Sha et al., 2020*) and defined nucleosome substrates ± accompanying acetylation (H3K9acK-14acK18ac; hereafter H3triᵃᶜ) (see **Methods**). In an endpoint assay at constant enzyme and substrate concentrations (and [*methyl*-³H]-SAM donor), we observed a significant increase in net methylation when the H3 tail was also acetylated (*Figure 2A*). As expected, methylation by MLL1C sequentially decreased towards H3K4 mono-, di-, and tri-methylated nucleosomal substrates, being undetectable on H3K4me3 (which also confirmed MLL1C targeting of this specific residue). Despite this, methyl group incorporation to each H3K4 methyl state substrate was consistently enhanced by accompanying H3triᵃᶜ (*Figure 2A*). Of note, we also tested the viability of methyllysine analogs (MLAs) (*Simon and Shokat, 2012*) ± H3triᵃᶜ as MLL1C substrates and observed no activity, indicating their unsuitability for such studies (*Figure 2—figure supplement 1D–E*).

We next measured the steady-state methylation kinetics of MLL1C towards nucleosomes with each H3K4 methyl state ± H3triᵃᶜ, and again observed that acetylation increased methyltransferase activity (*Figure 2B*). Using an extra sum-of-squares F-test, methylation for the H3triᵃᶜ nucleosomes was indicative of positive cooperativity because of better fit (p=0.0216) to the Hill equation (*Weiss, 1997*) (compare *Figure 2B* and *Figure 2—figure supplement 1C*; *Table 1* and *Supplementary file 1*). Because of the low level of enzymatic activity towards the unacetylated nucleosomes we could not make a statistically significant comparison between the Hill and Michaelis-Menten fits. There have been limited studies of MLL1C activity on nucleosomes (*Park et al., 2019*; *Patel et al., 2011*; *Xue et al., 2019*), so an overlooked potential allostery is understandable given the many possible interactions between this enzyme complex and substrate (*Lee et al., 2021*; *Park et al., 2019*).

On examining enzymatic parameters in detail, we noted that although overall $k_{cat}$ was ~17 fold greater for homotypic H3triᵃᶜ (over unmodified, H3 • H3) nucleosomes, the $K_{0.5}$ values (substrate concentration at half-maximal velocity/half-saturation for an allosterically regulated enzyme) were indistinguishable (*Table 1*). Therefore, although MLL1C catalytic efficiency toward H3triᵃᶜ nucleosomes was enhanced by an increase in $k_{cat}$, this catalytic efficiency was not driven by $K_{0.5}$, which suggested no increase in relative binding affinity. To further examine this, we returned to the dCypher approach to examine potential binding between the MLL1C query and a selection of nucleosome targets: H3 • H3, H3NΔ32 (lacking the first 32 residues of H3), Tailless (trypsin-digested nucleosomes to remove N- and C-terminal histone tails), and H3triᵃᶜ. At 50 mM NaCl, we observed no compelling difference in MLL1C binding to any of these targets (*Figure 2C* and *Figure 2—figure supplement 1E*). This is agreed with structural studies where binding between MLL1C and the nucleosome occurs primarily through interactions with DNA and, to a lesser degree, the H4 N-terminal tail (*Lee et al., 2021*; *Park et al., 2019*). Thus, the increased H3K4 methylation observed when the H3 N-terminal tail was acetylated is not due to enhanced MLL1C-nucleosome binding. Instead, H3 acetylation likely released the histone tail from the nucleosome, thereby increasing the apparent H3K4 concentration for MLL1C and enhancing methylation.

To definitively explore if the MLL1C methylation of H3K4 in H3triᵃᶜ nucleosomes was responding to *cis* and/or *trans* tail acetylation, we synthesized heterotypic substrates with only one H3K4 residue available in the *cis* or *trans* H3ac context (i.e. [H3triᵃᶜ • H3K4me3] *vs.* [H3 • H3tetraᵃᶜ]). Using these

substrates with MLL1C, we observed >threefold enhanced methylation (over [H3 • H3] or [H3 • H3K4me3]) when H3 tail acetylation was *cis* but no significant impact when *trans* (*Figure 2D*). This would support a model where H3 acetylation on the same tail releases H3K4 for methylation by MLL1C and appear to exclude a significant contribution from *trans*-tail mechanisms.

## Cellular level of H3K4 methylation is coupled to H3 N-terminal tail hyperacetylation

The above data suggested a molecular model for how the H3 N-terminal tail, via *cis* acetylation, becomes available for H3K4 reader binding or enzymatic modification in vitro. To determine if such acetylation could function as an accessibility switch in vivo, we developed a novel targeted middle-down mass spectrometry method to provide a single molecule quantitative measure of histone tail modification. We applied this method to acid-extracted histones from asynchronous MCF-7 breast cancer cells to measure the relationship between H3K4 methylation and tail acetylation on the same H3 proteoforms (*Holt et al., 2021*; *Smith et al., 2013*). As expected (*Garcia et al., 2007*; *Peach et al., 2012*; *Young et al., 2009*), the absolute amounts of H3K4me3 and higher Kac states (3ac,

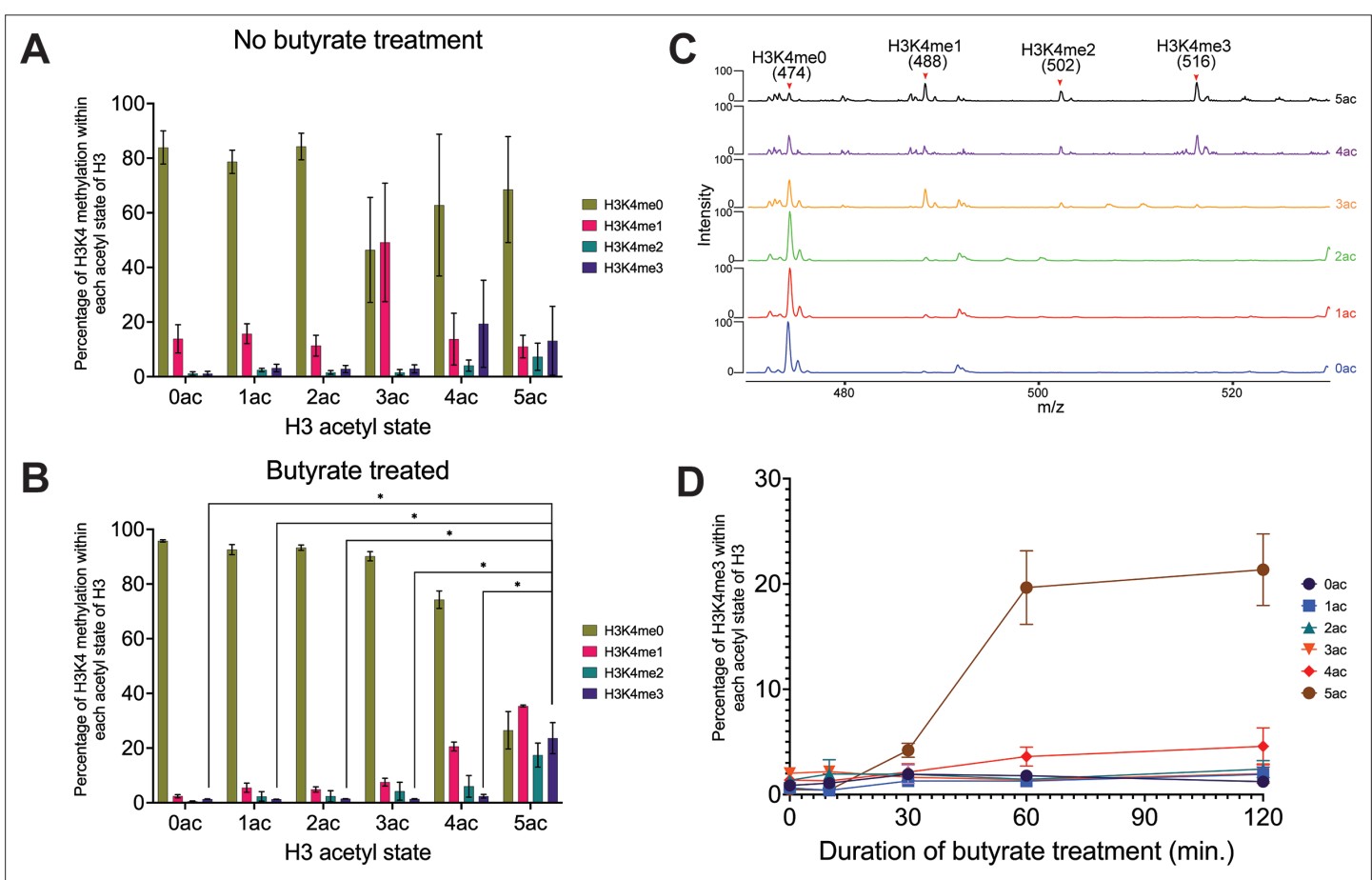

**Figure 3.** Middle-down mass spectrometric (MS) analysis reveals a hierarchical dependence between H3K4 methylation and *cis* H3 acetylation in MCF-7 cells. (**A**) Occupancy of H3K4 methyl states (me0-1-2-3) within each H3 acetyl state (0ac to 5ac) in asynchronous MCF-7 cells. (**B**) Occupancy of H3K4 methyl states within each H3 acetyl state after butyrate treatment (60 min). Asterisks represent <i>p-values <0.05. (**C**) Representative tandem mass spectra of the targeted C4$^{+1}$ fragment ion series (474 *m/z* unmodified; 488 *m/z* me1; 502 *m/z* me1; 516 *m/z* me3). Each spectrum is an average of MS2 spectra of the indicated H3 acetyl states after 60 min of butyrate treatment. K4 occupancy stoichiometry is directly correlated with H3 acetylation state and the targeted MS approach provides excellent signal-to-noise for confident quantitation. (**D**) Time course of H3K4me3 accumulation with respect to each H3 acetyl state after butyrate treatment. See **Methods** for further information on data acquisition and analysis. All data were collected in biological triplicate with error bars representing SEM.

The online version of this article includes the following figure supplement(s) for figure 3:

**Figure supplement 1.** H3 acetylation states with sodium butyrate treatment.

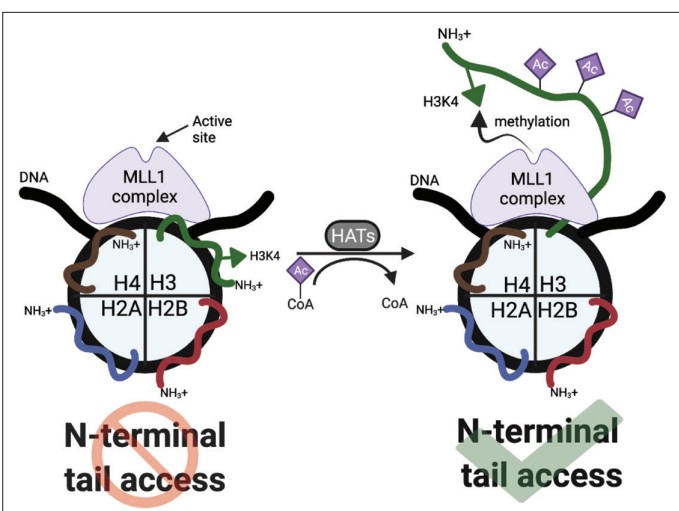

**Figure 4.** Regulation of H3K4 methylation by *cis*-tail H3 acetylation. Nucleosomal DNA is represented in black; each histone is labeled, with the core histone N-terminal tails colored to distinguish. MLL1 complex (MLL1C) is in purple.

4ac, and 5ac) across adjacent lysine residues were extremely low (*Figure 3—figure supplement 1A*). Nonetheless, H3K4me3 (<1% of total H3) was strictly associated with molecules that contained multiple acetylations (also <1% of total H3; *Figure 3A*). Given this relationship, we next addressed the hypothesis that increased lysine acetylation may release the H3 tail for more effective H3K4 methylation (i.e. acetylation precedes methylation). We treated MCF-7 cells with the KDAC inhibitor sodium butyrate and collected samples at multiple time points to measure the levels of H3K4 methylation with *cis* acetylation. H3 poly-acetylation rapidly increased upon butyrate treatment (*Figure 3—figure supplement 1B* and *Supplementary file 2*), as expected (*Holt et al., 2019*; *Young et al., 2009*). We observed an increase in all H3K4 methyl states relative to the unmodified state concomitant with increasing states of H3 acetylation (*Figure 3B–C* and *Supplementary file 2*), while H3K4me3 levels most dramatically increased in tandem with the 5ac H3 state (*Figure 3D*). An example of tandem mass spectra at each acetyl degree, showing the C4$^{+1}$ ion series from which K4 stoichiometry is measured, is shown in *Figure 3C*. These findings support a direct link, where acetylation releases nucleosome-bound H3 tails to localized H3K4 methyltransferases for subsequent methylation (*Figure 4*).

To explore whether other H3 lysine methyl marks are similarly affected by H3 acetylation, we examined the co-occurrence of H3K9me1-me2-me3 with increasing H3 acetyl states ± butyrate treatment in asynchronous HEK-293 cells (*Figure 3—figure supplement 1C–D*; *Supplementary file 3*). In contrast to findings with H3K4me3, H3K9me3 did not accumulate with hyperacetylated H3 proteoforms (i.e. 3ac, 4ac, and 5ac). Recent in vitro studies showed tested H3K9 methyltransferases have increased activity to partially acetylated substrates (*Trush et al., 2022*), so our MS data would suggest this is not a common in vivo mechanism: explanations could include that K9 was generally acetylated (and thus occupied) during histone tail release (all penta-acetylated (5ac) H3 proteoforms consisted of acetylation at H3K9, K14, and K18 at minimum (*Figure 3*, *Figure 3—figure supplement 1E* and *Supplementary file 4*)); or that H3K9 methyltransferases, in contrast to H3K4 methyltransferases, were not generally localized to regions that will accumulate butyrate-induced hyperacetylation.

Taken together, these findings demonstrate a specific *cis*-dependent H3ac regulatory switch that functions to control H3K4 methylation output (*Figure 4*). We posit that such a 'chromatin-switch' is important to the ability of cells to translate short-term acetylation signals at gene promoters to longer-term heritable marks of epigenetic memory (*Greer et al., 2014*; *Hörmanseder et al., 2017*; *Muramoto et al., 2010*; *Ng et al., 2003*).

## Discussion

While previous investigations identified a link between H3K4 methylation and H3 acetylation in diverse species (*Garcia et al., 2007*; *Nightingale et al., 2007*; *Strahl et al., 1999*; *Taverna et al., 2007*;

*Young et al., 2009*) the molecular basis for this link was unknown, and we posit the H3 acetyl 'chromatin switch' defined herein is conserved across eukaryotes. Our new understanding of the dynamic structure of nucleosome histone tails, alternating between collapsed (i.e. nucleosome-bound) and accessible forms (*Marunde et al., 2022a*; *Morrison et al., 2018*), has made more plausible the notion that tail availability could be driven by combinatorial *cis* acetylation to directly promote H3K4 methylation. Histone tail lysine acetylation (Kac) can directly recruit residue-specific readers, e.g., bromodomains (*Musselman et al., 2012*), but acetylation also neutralizes the positive charge on lysine residues and relieves their interaction with negatively charged DNA (i.e. altering the histone tail-DNA binding equilibrium) (*Marunde et al., 2022a*; *Morrison et al., 2018*). In the nucleosome context, this decreased histone-DNA binding supports the increased engagement of reader domains that have no direct affinity for the Kac. Conversely, isolated histone tail peptides are 'constitutively open' (no DNA to engage), and thus not subject to this mode of regulation.

In this study, we showed that hyperacetylation of the H3 N-terminal tail promoted rapid accumulation of H3K4 methylation in cis, most likely by increasing the availability of the substrate residue to the MLL1C active site (*Figure 4*). This finding was supported by our in vitro enzymatic and in vivo mass spectrometric analyses. Methylation assays with MLL1C revealed significantly enhanced enzyme activity towards nucleosome substrates with co-incident acetylated (H3tri$^{ac}$) over unmodified H3 (*Figure 2A–B*; *Table 1*). dCypher assays demonstrated that the acetylation-mediated increase in H3K4 methylation does not involve stabilized interactions between MLL1C and the nucleosome (*Figure 2D*). And providing major insight, heterotypic nucleosomes containing two distinct PTM-defined forms of histone H3 definitively showed that MLL1C activity was only enhanced in a *cis* H3 N-terminal triacetylation context (*Figure 2D*). In agreement with these in vitro findings, middle-down mass spectrometry showed that H3 hyperacetylation and H3K4 methylation co-occurred on the same histone tails in actively cycling cells; furthermore, upon butyrate treatment, H3K4 methylation increased for the most highly acetylated proteoforms (*Figure 3*). The in vivo relationship required higher degrees of acetylation (preferring at least four acetyl groups per molecule: e.g. H3K9acK14acK18acK23acK27ac) for the most effective conversion to the H3K4me3 state. This could be a function of yet to be explained in vivo acetylation hierarchies by KATs that are outside the scope of this study but will be important to resolve. Together with binding studies that identify the positive impact of *cis* H3 tail hyperacetylation on H3K4 reader engagement (*Figure 1* and *Figure 1—figure supplement 1*; *Marunde et al., 2022a*; *Morrison et al., 2018*), our findings suggest a molecular switch that governs when and where the histone H3 N-terminus is available for H3K4-related transactions. Such a switch could be used to establish and heritably maintain the location and function of transcriptional promoters across the genome.

A continued observation from this study is that the binding preference of readers (in this case PHD-fingers that engage histone H3K4) narrows on nucleosomes relative to histone peptides (*Figure 1* and *Figure 1—figure supplement 1*; *Marunde et al., 2022a*; *Morgan et al., 2021*). Such differences highlight the importance of using a more representative target to identify the most likely/physiologically relevant interactions. However, we also add to the literature confirming the importance of a nucleosome substrate for enzymatic studies (*Strelow et al., 2016*; *Stützer et al., 2016*). In this regard our steady-state kinetic methylation assays with MLL1C revealed an intriguing (and previously undescribed) positive cooperativity with its preferred nucleosome substrates (*Figure 2B vs. Figure 2—figure supplement 1C*). It is important to consider the suggestion that allosteric factors could regulate interactions between MLL1C and the nucleosome, especially since previous kinetic analyses of this [enzyme: substrate] pair did not address such behavior (*Park et al., 2019*; *Xue et al., 2019*). However, positive cooperativity was not evident at the level of [MLL1C: nucleosome] binding, which was independent of substrate identity (i.e. ±H3tri$^{ac}$) at ionic conditions similar to the catalytic assay (*Figure 2C*). Still to be determined are the signals that drive and regulate this cooperativity and its function in MLL1-catalyzed methylation, especially in a higher-order chromatin context.

Taken together, our study highlights a previously unrecognized regulatory mechanism for how writers might engage the histone H3 tail in vivo. Although this work focused on H3K4, it will be important to ascertain the consequence of acetylation (or acylation)-mediated changes in accessibility and the function of modifiers and readers of the other lysines on H3, as well as on the other core histones (H2A, H2B, and H4). Underscoring the need for such studies, we note recent in vitro analyses employing unmodified nucleosomes sequentially targeted by purified KATs and KMTs suggesting

that the acetylation landscape can impact multiple methyltransferases (*Trush et al., 2022*). While that study reported enhanced G9a activity towards H3K9 as a function of p300-mediated histone acetylation, our mass spectrometric analysis from cells did not show a connection between H3K9me3 and H3 hyperacetylation (*Figure 3—figure supplement 1C–D*). The field will require a more detailed analysis of the in vivo contributions of various KATs/KDACs to regulate histone tail accessibility (in *cis* and/or *trans*) for other chromatin-modifying enzymes to further uncover the molecular details of any sequential histone code.

Finally, we note that multiple studies have identified H3K4 methylation and H3 acetylation as active marks because of their co-occupancy on the promoters and gene bodies of transcribed genes (*Santos-Rosa et al., 2002*; *Strahl et al., 1999*; *Wozniak and Strahl, 2014*). Our findings agree with these observations, but also uncover a previously unrecognized mechanism of H3 cross-talk that impinges on fundamental functions of H3 acetylation and K4 methylation in gene regulation. While this work shows the impact of *cis* tail H3 acetylation on the H3K4 writer MLL1, we note a companion study highlighting how the resulting PTM signature (*cis >> trans*-H3K4me3tri[ac]) promotes BPTF-nucleosome engagement (*Marunde et al., 2022a*). Given its central importance, we predict this mechanism may be a target for dysregulation in human disease.

## Methods

### Expression and purification of GST-tagged PHD reader proteins

Expression constructs for GST-tagged PHD domains from KDM7A (Uniprot #Q6ZMT4; residues 1–100), DIDO1 (Uniprot #Q9BTC0; residues 250–340), and MLL5 (Uniprot #Q8IZD2; residues 100–180) were synthesized in a pGEX-4T-1 vector (*BioMatik Corporation*) and provided by Dr. Mark T. Bedford (UT MD Anderson Cancer Center). Recombinant proteins were expressed and purified as described (*Jain et al., 2020*). See *Supplementary file 5* for details about the constructs.

### Expression, purification, and assembly of the MLL1 core complex (MLL1C)

Methods for the expression, purification, and assembly of the MLL1 core complex (MLL1C: MLL1 SET domain, WDR5, RbBP5, Ash2L, and DPY30) were adapted from published protocols (*Usher et al., 2021*). A polycistronic recombinant expression construct containing the MLL1 SET domain (Uniprot Q03164; residues 3745–3969), WDR5 (Uniprot P61964; residues 2–334), RbBP5 (Uniprot Q15291; residues 1–538), and Ash2L (Uniprot Q9UBL3-3; residues 1–534) in pST44 vector (*aka.* MWRA construct) was a kind gift from Dr. Song Tan (*Tan et al., 2005*). WDR5 was cloned with an N-terminal hexahistidine (6HIS) tag and TEV protease site to enable purification by immobilized metal affinity chromatography (IMAC), and tag removal by enzymatic cleavage. See *Supplementary file 5* for details about the constructs. Rosetta pLysS *E. coli* cells were transformed with the plasmid and grown on LB plates with 50 µg/mL carbenicillin. Single colonies were used to inoculate 50 mL starter cultures of Terrific Broth (TB) containing 50 µg/mL carbenicillin and grown at 37 °C for 16 hr. This culture was transferred to 1 L of TB+ carbenicillin and grown to $OD_{600}$ ~0.6 (37°C, 200 RPM, ~4 hr). Recombinant protein expression was induced with 1 mM Isopropyl β-D-1-thiogalactopyranoside (IPTG, *Sigma*; 16 °C, 200 RPM, 20 hr). Cells were harvested by centrifugation (5000 RPM, 4 °C) and flash frozen in liquid nitrogen.

Frozen cell pellets were resuspended in 50 mL of lysis buffer (50 mM Tris-HCl pH 7.5, 300 mM NaCl, 30 mM imidazole, 3 mM dithiothreitol, and 1 µM $ZnCl_2$) containing 0.5 mg/mL lysozyme (*Sigma*), 250 U Pierce Universal Nuclease (*Thermo Fisher*), and an EDTA-free protease inhibitor cocktail (*Roche*) and rotated at 4 °C for 1 hr. The resultant mixture was then sonicated [five cycles of 30 s on/30 s off at 50% output] and centrifuged at 4 °C, 15,000 RPM for 35 min. The clarified lysate has flowed over a 5 mL HisTrap nickel column (*Cytiva*) using an AKTA Pure FPLC (*Cytiva*) at 0.5 mL/min; all FPLC steps were conducted at 4 °C. Unbound molecules were removed with 20 column volumes of wash buffer (WB: 50 mM Tris-HCl, pH 7.5, 300 mM NaCl, 30 mM imidazole, 3 mM DTT, and 1 µM $ZnCl_2$ at 2 mL/min). The 6HIS-tagged MWRA was eluted in a 15-column volume linear gradient from WB to Elution Buffer (WB+ 500 mM imidazole) at 2 mL/min. Fractions containing 6HIS-tagged MWRA were identified by SDS-PAGE, pooled, and supplemented with 6HIS-tagged TEV protease (purified as described: *Nautiyal and Kuroda, 2018*) at a 1:100 enzyme to substrate molar ratio to cleave the 6HIS-tag on WDR5. This mixture was dialyzed against three changes of WB (each 2 L for at least 4 hr at 4 °C)

and 6HIS-TEV removed from cleaved MWRA via IMAC (*Usher et al., 2021*). MWRA flow-through protein solution was concentrated to ~15 mL using a 30 kDa MWCO centrifugal filter (*EMD Millipore*), ensuring not to concentrate to where the solution became yellow/cloudy and viscous. MWRA complex was resolved over a HiLoad 16/60 Superdex 200 pg gel filtration (GF) column (*Cytiva*) pre-equilibrated in 20 mM Tris-HCl, pH 7.5, 300 mM NaCl, 1 mM TCEP, and 1 μM ZnCl₂. Fractions containing stoichiometric MWRA sub-complex were identified by SDS-PAGE, pooled, and concentrated to ~15 mL.

HisDPY30 was expressed, purified, and cleaved to remove the 6HIS-tag as described (*Patel et al., 2009*; *Usher et al., 2021*). A twofold molar excess of DPY30 was added to the MWRA sub-complex and incubated on ice for 1 hr. Following incubation, the resulting MLL1C was isolated by gel filtration as described for MWRA, with fractions containing the stoichiometric complex identified by SDS-PAGE, pooled, concentrated to ~10 μM, and flash frozen. For dCypher experiments with MLL1C, the 6HIS-tag was retained on DPY30.

## Peptides

All peptides were synthesized at the UNC peptide synthesis core facility (RRID: SCR_017837), using Fmoc solid phase synthesis, on an automated peptide synthesizer (PTI Symphony or CEM Liberty Blue). The peptides were purified by preparative RP-HPLC and characterized by MALDI-TOF MS and analytical HPLC.

## PTM-defined nucleosomes

All PTM-defined nucleosomes (*Supplementary file 5*; homotypic unless stated otherwise) were from the dNuc or versaNuc portfolios (*EpiCypher*). PTMs were confirmed by mass-spectrometry and immunoblotting (if an antibody was available) (*Goswami et al., 2021*; *Marunde et al., 2022b*; *Marunde et al., 2022a*; *Weinberg et al., 2019*).

Nucleosomes with methyllysine analogs (MLA) at H3K4 were created by the versaNuc approach (*Marunde et al., 2022a*). In brief, histone H3 peptides (aa1-31; A29L) containing K4C and any PTMs of interest were site-specifically reacted with the corresponding haloalkylamine: (2-bromoethyl) trimethyl ammonium bromide for KCme3; 2-chloro-N,N-dimethyl-ethylamine hydrochloride for KCme2; or 2-chloroethyl(methyl)ammonium chloride for (KCme1) under SN₂ reaction conditions as previously (*Simon, 2010*; *Simon and Shokat, 2012*), and purified for individual ligation to an H3 tailless nucleosome precursor (H3.1NΔ32 assembled on 147 bp 5' biotinylated 601 DNA; #16–0016). The resulting nucleosomes (assembled at 50–100 μg scale) contained minimal free DNA (<5%), undetectable levels of peptide precursor, and ≥90% fully-defined full-length H3.1 (e.g. *Figure 2—figure supplement 1D–E*).

Heterotypic nucleosomes were assembled from PTM-defined histone octamers containing N-terminally bridged H3 dimers, and the bridge was removed in a scarless manner by approaches to be described elsewhere (Manuscript in Preparation). Heterotypic identity was confirmed at all synthesis steps by analyses additional to those used for homotypics, including Nuc-MS on representative final nucleosomes (*Schachner et al., 2021*). Heterotypic nomenclature describes each PTM-defined histone in the nucleosome, such that [H3K4me3K9acK14acK18ac • H3] *vs.* [H3K4me3 • H3K9acK14acK18ac] contain the same total PTM complement but distributed *cis* or *trans* on the H3 N-termini.

## dCypher assays

*dCypher* binding assays with PTM-defined nucleosomes were performed under standard conditions that titrate query (e.g. epitope-tagged reader domain(s)) to a fixed concentration of target (e.g. biotinylated PTM-defined nucleosome) with the appropriate Alpha Donor and Acceptor beads (*Perkin Elmer*) (*Jain et al., 2020*; *Marunde et al., 2022b*; *Weinberg et al., 2019*). Binding curves [query: target] were generated using a non-linear 4PL curve fit in Prism 9.0 (*GraphPad*). For each query, the relative EC₅₀ ($EC_{50}^{rel}$) and hillslope values were derived from the best binding target. $EC_{50}^{rel}$ is the half-maximal signal for the specified target. Where necessary, we excluded query concentration values determined to be beyond a query's hook point (signal inhibition due to query exceeding bead saturation). The $EC_{80}^{rel}$ was selected as the optimal probing concentration for discovery screens because of the robust signal-to-background and to provide the best opportunity to bind targets without saturating the primary target signal. To compute $EC_{80}^{rel}$ values, we used the formula $EC_F^{rel} = (F/(100 - F))^{1/H} \times EC_{50}^{rel}$; $F=80$; and H=hillslope.

Briefly, 5 µL of GST-tagged reader domain was incubated with 5 µL of 10 nM biotinylated nucleosomes (e.g. *EpiCypher* #16–9001) for 30 min at room temperature in 20 mM HEPES pH 7.5, 250 mM NaCl, 0.01% BSA, 0.01% NP-40, 1 mM DTT in a 384-well plate. A mix of 10 uL of 2.5 µg/mL glutathione acceptor beads (*PerkinElmer*, AL109M) and 5 µg/mL streptavidin donor beads (*PerkinElmer*, 6760002) was prepared in 20 mM HEPES pH 7.5, 250 mM NaCl, 0.01% BSA, 0.01% NP-40 and added to each well. The plate was incubated at room temperature in subdued lighting for 60 min, and AlphaLISA signal was measured on a PerkinElmer 2104 EnVision (680 nm laser excitation, 570 nm emission filter ± 50 nm bandwidth). Each binding interaction was performed in duplicate in a 20 µL mix in 384 well plates.

MLL1C binding assays (*Figure 2C*) were performed as above except using Nickel-chelate acceptor beads (10 µg/mL; *Perkin Elmer* AL108M), streptavidin donor beads (20 µg/mL; *Perkin Elmer*) and modified assay buffer (20 mM Tris pH 7.5 + 50 mM NaCl, 0.01% BSA, 0.01% NP-40, and 1 mM DTT); [NaCl] was optimized via a titration assay and 50 mM chosen for subsequent analyses.

## In vitro methylation assays

Methylation assays (*Shinsky et al., 2015*) were performed at 15 °C for 3 hr using purified MLL1C enzyme and nucleosome substrate in a reaction volume of 20 µL [in 50 mM HEPES, pH 8.0, 1 mM DTT, 1 µM ZnCl$_2$; 10 µM of 9:1 *S*-adenosyl-L-methionine (SAM) *p*-toluenesulfonate salt (*Sigma*) to *S*-adenosyl-L-[*methyl*-$^3$H]-methionine ([*methyl*-$^3$H]-SAM) (*PerkinElmer*)]. Concentrations of MLL1C and NaCl were optimized from 2D-titration methylation assays at [0, 4, and 40 nM MLL1C] and [0, 50, and 300 mM NaCl] with 2 µg of the chicken oligo-nucleosome substrate (*Figure 2—figure supplement 1A–B*; *Morris et al., 2007*). It is notable that, across a range of concentrations, MLL1C stability decreases as temperature increases and methyltransferase activity has been reported to be enhanced in sub-physiological NaCl concentrations (*Namitz et al., 2019*; *Shinsky et al., 2015*). For endpoint methylation assays, 4 nM MLL1C and 100 nM nucleosome substrates were tested as above. For steady-state kinetics, 4 nM MLL1C was incubated with a nucleosome substrate titration (0, 23.4, 35.1, 52.7, 79, 119, 178, 267, and 400 nM) and reactions quenched with 5 µL of 5 X SDS loading dye. For steady-state kinetic assays, 0.5 µg bovine serum albumin was added to each reaction after quenching to act as a loading guide. To analyze methylation, quenched reactions were resolved by 15% Tris-Glycine SDS-PAGE. Gels were stained with Coomassie dye and bands containing mononucleosomes were excised (with serum albumin as a supporting lane marker) and incubated in a solution of 50% Solvable (*PerkinElmer*) and 50% water at 50 °C for 3 hr. Mixture and gel slices were then combined with 10 mL of Hionic-Fluor scintillation fluid (*PerkinElmer*), dark-adapted overnight, and radioactivity measured on a Liquid Scintillation Counter (*Beckman Coulter*).

## Middle-down mass spectrometry of MCF-7 and HEK-293 cells ± KDAC inhibition

MCF-7 breast cancer cells (ATCC HTB-22) were grown in MEM (*Gibco*) supplemented with 10% fetal bovine serum (*VWR*), 100 I.U. penicillin, 100 µg/mL streptomycin (*Corning*), 0.01 mg/mL human recombinant insulin (*Gibco*), and 5 µg/mL plasmocin (*Invivogen*) at 37 °C and 5% CO$_2$. HEK-293 (ATCC CRL-1573) was grown in DMEM (*Gibco*) supplemented with 10% fetal bovine serum (*Corning*), 100 I.U. penicillin, and 100 µg/mL streptomycin (*Gibco*) at 37 °C and 5% CO$_2$. Both cell lines were authenticated via STR profiling and confirmed to be *Mycoplasma* negative.

For mass spectrometric (MS) analysis ± KDAC inhibition, cells were cultured in 150 mm dishes to ~80% confluence and treated with 5 mM sodium butyrate (or equivalent volume of water) in triplicate for 0, 10, 20, 30, 60, 120 min. Cells were washed with cold PBS (11.9 mM phosphates, 137 mM NaCl, 2.7 mM KCl) to remove residual sodium butyrate, harvested by scraping, and flash frozen in liquid nitrogen. Histones were acid extracted after nuclei isolation as described (*Holt et al., 2021*). Isolated histones were resuspended in 85 µL 5% acetonitrile, 0.2% trifluoroacetic acid (TFA), and resolved by offline high-performance liquid chromatography (HPLC) as described (*Holt et al., 2021*). Reverse Phase HPLC fractionation was performed with a U3000 HPLC system (*Thermo Fisher*) with a 150 × 2.1–mm Vydac 218TP 3 µm C18 column (*HiChrom* # 218TP3215), at a flow rate of 0.2 mL/min using a linear gradient from 25 – 60% B in 60 min. The composition of buffers used was A: 5% acetonitrile and 0.2% TFA and B: 95% acetonitrile and 0.188% TFA. After chromatographic separation and fraction collection, histone H3.1 was selected, diluted in 100 mM ammonium acetate (pH = 4), and

digested with Glu-C protease (*Roche*) at 10:1 protein:enzyme for 1 hr at room temperature prior to mass spectrometric analysis.

The digested samples were diluted to 2 µg/µL. Online HPLC was performed on a U3000 RSLC nano Pro-flow system using a C3 column (Zorbax 300 SB-C3, 5 µm; *Agilent*). Samples were maintained at 4 °C and 1 µL injected for each analysis using a microliter pickup. A linear 70 min gradient of 4–15% B was used (Buffer A: 2% acetonitrile, 0.1% formic acid and Buffer B: 98% acetonitrile, and 0.1% formic acid) with a flow rate of 0.2 µL/min. The column eluant was introduced into an Orbitrap Fusion Lumos mass spectrometer (*Thermo Fisher*) by nano-electrospray ionization. A static spray voltage of 1900 V and an ion transfer tube temperature of 320 °C were set for the source.

A Fusion Lumos mass spectrometer was used to generate MS data. The 9th charge state of histone H3 was targeted for analysis. MS1 analysis was acquired in the orbitrap with a normal mass range and a 60 k resolution setting in positive ion mode. An Automatic Gain Control (AGC) target of 5.0E5 with a 200 ms maximum injection time, three microscans, and a scan range of 585–640 *m/z* was used to identify desired ions for fragmentation. MS2 acquisition was performed in both orbitrap and ion trap modes. Both modes used electron transfer dissociation (ETD), a reaction time of 18ms, and an injection time of 200 ms. A normal scan range was used for the orbitrap mode with a resolution setting of 30 k and an AGC target of 5.0E5, with two micro scans. A narrow scan range of 470–530 *m/z*, targeting ions indicative of K4 modification states, was used for the ion trap mode MS2 with an ACG target of 3.0E4, quadrupole isolation, maximum injection time of 100ms, and eight microscans.

These MS methods were used with two technical replicates per biological replicate (n=3). An MS ion trap mode with a targeted mass list was used to increase sensitivity to identify known low-abundance K4me3-containing proteoforms. The ion trap MS2 spectra were averaged for each H3 acetyl state based on known retention times, and the intensities of ions indicative of the K4 methylation states were manually recorded. Retention times for each acetyl state were approximated as 0ac 35–40 min, 1ac 45–55 min, 2ac 55–60 min, 3ac 62–68 min, 4ac 69–73 min, and 5ac 74–78 min. Precursor mass was used as an additional confirmation and filter of the correct acetyl state. For scans yielding low signal and high noise (i.e. 5ac at 0 min butyrate treatment), data were manually curated before averaging. Within acetyl states, the relative proportion of fragment ions for unmodified, mono-, di-, and tri-methylation of the H3K4 ion at respective *m/z* of 474, 488, 502, and 516 were recorded per MS run. Values were averaged across replicates of the same conditions and normalized to one hundred percent. A two-tailed t-test was used for significance. Raw MS data is available at (ftp://massive.ucsd.edu/MSV000089089/, ftp://massive.ucsd.edu/MSV000091578/).

The MS method used here is highly targeted to most effectively address the mechanistic or single molecule link between H3 acetylation degree and H3K4 occupancy. The strategy used prioritizes the optimization of efficient selection of acetyl degree and of the signal-to-noise for the $C4^{+1}$ ion series. This is to the exclusion of other information that can typically be derived from untargeted approaches. For example, because trapping mass spectrometers are limited in the number of ions, we dispose of unnecessary ions to the gas phase to enrich the C4 ions series. This method provides a direct measure of the stoichiometry of K4un, K4me1, K4me2, and K4me3 within an acetyl state.

## Acknowledgements

KJ is supported by a Postdoctoral Training Fellowship from the National Institutes of Health (NIH; T32CA217824) to the UNC Lineberger Cancer Center and a Postdoctoral Fellowship from the American Cancer Society (PF-20-149-01-DMC). This work was also supported by NIH grants to NLY (R01GM139295, P01AG066606, and R01CA193235), MSC (R01CA140522), *EpiCypher* (R43CA236474, R44GM117683, R44CA214076, and R44GM116584), and to BDS (R35GM126900). We thank colleagues for the generous supply of materials (see Methods) and members of the Cosgrove, Young, *EpiCypher,* and Strahl labs for helpful discussions and suggestions.

## Additional information

### Competing interests

Matthew R Marunde, Jonathan M Burg, Susan L Gloor, Irina K Popova, Nathan W Hall, Anup Vaidya, Sarah A Howard, Hailey F Taylor, Laylo Mukhsinova, Ugochi C Onuoha, Emily F Patteson, Ellen N Weinzapfel, Marcus A Cheek, Matthew J Meiners, Martis W Cowles, Zu-Wen Sun: is affiliated with EpiCypher, Inc The author has no financial interests to declare. Zachary B Gillespie, Keli L Rodriguez: is affiliated with EpiCypher, Inc. The author has no financial interests to declare. Krzysztof Krajewski: owns options in EpiCypher, Inc. Michael S Cosgrove: owns stock/serves on the Consultant Advisory Board for Kathera Bioscience Inc and holds266 US patents (8,133,690; 8,715,678; and 10,392,423) for compounds/methods for inhibiting267 SET1/MLL family complexes. Michael-Christopher Keogh: is a BOD member of EpiCypher Inc. Brian D Strahl: is a co-founder and BOD member of EpiCypher, Inc. The other authors declare that no competing interests exist.

### Funding

| Funder | Grant reference number | Author |
| --- | --- | --- |
| American Cancer Society | PF-20-149-01-DMC | Kanishk Jain |
| National Institute of General Medical Sciences | R35GM126900 | Brian D Strahl |
| National Institute of General Medical Sciences | R01GM139295 | Nicolas L Young |
| National Cancer Institute | T32CA217824 | Kanishk Jain |
| National Institute of General Medical Sciences | P01AG066606 | Nicolas L Young |
| National Cancer Institute | R01CA193235 | Nicolas L Young |
| National Cancer Institute | R01CA140522 | Michael S Cosgrove |
| National Cancer Institute | R43CA236474 | Michael-Christopher Keogh |
| National Institute of General Medical Sciences | R44GM117683 | Zu-Wen Sun |
| National Cancer Institute | R44CA214076 | Michael-Christopher Keogh |
| National Institute of General Medical Sciences | R44GM116584 | Zu-Wen Sun |

The funders had no role in study design, data collection and interpretation, or the decision to submit the work for publication.

### Author contributions

Kanishk Jain, Conceptualization, Data curation, Formal analysis, Investigation, Visualization, Methodology, Writing – original draft, Writing – review and editing; Matthew R Marunde, Conceptualization, Data curation, Formal analysis, Investigation, Writing – review and editing; Jonathan M Burg, Conceptualization, Resources, Data curation, Formal analysis, Methodology, Writing – review and editing; Susan L Gloor, Data curation, Methodology, Writing – review and editing; Faith M Joseph, Data curation, Formal analysis, Methodology, Writing – review and editing; Karl F Poncha, Data curation, Formal analysis, Writing – review and editing; Zachary B Gillespie, Keli L Rodriguez, Irina K Popova, Nathan W Hall, Anup Vaidya, Sarah A Howard, Hailey F Taylor, Ugochi C Onuoha, Emily F Patteson, Spencer W Cooke, Bethany C Taylor, Ellen N Weinzapfel, Marcus A Cheek, Matthew J Meiners, Data curation, Writing – review and editing; Laylo Mukhsinova, Geoffrey C Fox, Data curation; Kevin EW Namitz, Resources, Methodology, Writing – review and editing; Martis W Cowles, Project administration, Writing – review and editing; Krzysztof Krajewski, Michael S Cosgrove, Resources, Writing – review and editing; Zu-Wen Sun, Project administration; Nicolas L Young, Conceptualization, Data curation, Formal analysis, Methodology, Project administration, Writing – review and editing; Michael-Christopher

Keogh, Conceptualization, Resources, Formal analysis, Supervision, Project administration, Writing – review and editing; Brian D Strahl, Conceptualization, Formal analysis, Supervision, Funding acquisition, Writing – original draft, Project administration, Writing – review and editing

### Author ORCIDs
Kanishk Jain  http://orcid.org/0000-0001-8039-0464
Geoffrey C Fox  http://orcid.org/0000-0001-5898-0847
Kevin EW Namitz  http://orcid.org/0000-0003-4434-1475
Nicolas L Young  http://orcid.org/0000-0002-3323-2815
Michael-Christopher Keogh  http://orcid.org/0000-0002-2219-8623
Brian D Strahl  http://orcid.org/0000-0002-4947-6259

### Decision letter and Author response
Decision letter https://doi.org/10.7554/eLife.82596.sa1
Author response https://doi.org/10.7554/eLife.82596.sa2

---

## Additional files

### Supplementary files
• Supplementary file 1. Steady-state Michaelis-Menten kinetic parameters. [a] = methylation signal observed was indistinguishable from the background and thus kinetic parameters could not be determined.

• Supplementary file 2. Mass Spectrometric data for H3K4 methylation and H3 acetylation in asynchronous MCF-7 cells as a function of butyrate treatment. (See Excel file).

• Supplementary file 3. Mass Spectrometric data for H3K9 methylation and H3 acetylation in asynchronous HEK-293 cells as a function of butyrate treatment. (See Excel file).

• Supplementary file 4. Mass Spectrometric data for H3K9acK14acK18ac abundance within H3 acetyl states in asynchronous HEK-293 cells as a function of butyrate treatment. (See Excel file).

• Supplementary file 5. A table resource detailing recombinant protein constructs, peptides, and nucleosomes used in this study. (See Excel file).

• MDAR checklist

### Data availability
Raw MS data is publicly available and has been uploaded to the UCSD MassIVE database (MSV000089089 and MSV000091578). All analyzed data are reported in the manuscript and Supporting Files.

The following datasets were generated:

| Author(s) | Year | Dataset title | Dataset URL | Database and Identifier |
|---|---|---|---|---|
| Jain K | 2022 | An acetylation-mediated chromatin switch governs H3K4 methylation read-write capability | https://massive.ucsd.edu/ProteoSAFe/dataset.jsp?accession=MSV000089089 | MassIVE, MSV000089089 |
| Jain K | 2023 | An acetylation-mediated chromatin switch governs H3K4 methylation read-write capability | https://massive.ucsd.edu/ProteoSAFe/dataset.jsp?accession=MSV000091578 | MassIVE, MSV000091578 |

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
