## [Editor Report]

This is a fundamental study revealing that cis H3 tail acetylation promotes nucleosome accessibility to histone methyl readers and writers. The findings are supported by compelling evidence providing an explanation for long-observed mechanisms of chromatin crosstalk.

---

## [Decision Letter]

**Decision letter after peer review:**

Thank you for submitting your article "An acetylation-mediated chromatin switch governs H3K4 methylation read-write capability" for consideration by *eLife*. Your article has been reviewed by 2 peer reviewers, one of whom is a member of our Board of Reviewing Editors, and the evaluation has been overseen by and Jessica Tyler as the Senior Editor. The reviewers have opted to remain anonymous.

The two reviewers have discussed their reviews with one another, and both agreed that all the comments should be experimentally addressed before we can consider publication at *eLife*. We hope you find these comments helpful for your revision.

*Reviewer #1 (Recommendations for the authors):*

1) The authors propose that cis acetylation affects MLL1C activity on H3K4. Experiments are needed to compare the effect of cis vs trans acetylation of histone H3 tail (using asymmetrically acetylated nucleosome) on H3K4 methylation.

2) If acetylation regulates enzymatic activity through tail accessibility, such an effect can also been observed on H3K9 methyltransferases. This can be tested using the dCypher nucleosomes and G9a or Suv39H enzymes. Also, are increased H3K9 methylation with 4ac or 5ac observed in the HDACi treated MCF-7 cells?

3) The MS data in MCF-7 cells reveal that H3K4 methylation is drastically increased only in the 5-ac but not 3-ac histones. This is inconsistent with the in vitro biochemical data. Experiments are needed to compare the difference between H3 with 5-ac and 3-ac in the in vitro enzymatic assays.

---

## [Author Response]

Reviewer #1 (Recommendations for the authors):1) The authors propose that cis acetylation affects MLL1C activity on H3K4. Experiments are needed to compare the effect of cis vs trans acetylation of histone H3 tail (using asymmetrically acetylated nucleosome) on H3K4 methylation.

We thank the reviewer for bringing this important point to our attention. To address this, we expended considerable resources to create new fully PTM-defined heterotypic (to accompany our homotypic) nucleosomes (note nomenclature to minimize confusion with asymmetric/symmetric DNA methylation) to directly test whether MLL1’s activity enhancement in the context of H3 tail acetylation occurs in *cis* or in *trans*. As shown in Figure 2D, enhancement of H3K4 methylation only occurs with heterotypic nucleosomes that have an available H3K4 residue with tail acetylation in *cis* (H3K4me3 • H3K9acK14acK18ac (hereafter H3tri^ac^)) and is not seen in H3K4 methylatable nucleosomes with tail acetylation in *trans* (H3 • H3K4acK9acK14acK18ac (hereafter H3tetra^ac^)). These exciting new findings greatly strengthen our study and provide more definitive mechanistic details of H3ac → H3K4me regulation.

2) If acetylation regulates enzymatic activity through tail accessibility, such an effect can also been observed on H3K9 methyltransferases. This can be tested using the dCypher nucleosomes and G9a or Suv39H enzymes. Also, are increased H3K9 methylation with 4ac or 5ac observed in the HDACi treated MCF-7 cells?

In the revised Discussion we have made clearer reference to a recent study by Trush et al. (10.1016/j.bbagrm.2022.194845) reporting the enhanced activity of H3K9 methyltransferases (G9a and SUV39H) when nucleosomal substrates are in vitro acetylated with KATs. However, our mass spectrometry analyses do not find a direct correlation between H3 acetylation levels and H3K9 methylation in cells (Figure 3 —figure supplement 1C-D). It is important to also note that the 5ac state almost exclusively contains K9ac, which blocks any methyltransferase activity at this site and the 4ac state is also dramatically enriched for K9ac. We take these data to mean that in vivo hyperacetylated H3 could function, at least in part, to prevent rather than stimulate repressive H3K9 methylation. However, we note there are certainly enzymes that can access the tail without acetylation, notably the acetyltransferases. Thus, these and other findings suggest not all enzymes are equally affected by H3ac accessibility. Per the reviewer’s suggestion, we have included our MS H3K9 methylation results and mention the possibility of other enzymes being influenced by KATs in the discussion (including the recent Trush paper).

3) The MS data in MCF-7 cells reveal that H3K4 methylation is drastically increased only in the 5-ac but not 3-ac histones. This is inconsistent with the in vitro biochemical data. Experiments are needed to compare the difference between H3 with 5-ac and 3-ac in the in vitro enzymatic assays.

The reviewer raises an intriguing point between our in vitro studies and the H3ac signature associated with H3K4me3 in cells. Intriguingly, our MS analyses reveal that H3ac proteoforms used in our in vitro assays (K9acK14acK18ac combination) exist in vivo almost exclusively within the 4ac and 5ac states (Figure 3 – —figure supplement 1E). Butyrate treatment primarily impacts how much of the K9acK14acK18ac combination is present within 4ac vs. 5ac, skipping the 3ac state. Nonetheless, this 3ac state is sufficient to direct the *cis*-effect on MLL1C mediated methylation. While it is reasonable to ask that we compare a 5ac nucleosome substrate versus the 3ac, we needed to prioritize our efforts and use the limited time and resources to develop heterotypic nucleosomes. We felt addressing the *cis* vs. *trans* question (asked by both reviewers) would have much a greater impact on the study.